# In Vitro Assessment of Antimicrobial Activity of Phytobiotics Composition towards of Avian Pathogenic *Escherichia coli* (APEC) and Other *E. coli* Strains Isolated from Broiler Chickens

**DOI:** 10.3390/antibiotics11121818

**Published:** 2022-12-15

**Authors:** Karolina A. Chodkowska, Hubert Iwiński, Karolina Wódz, Tomasz Nowak, Henryk Różański

**Affiliations:** 1Krzyżanowski Partners Spółka z o.o., Zakładowa 7, 26-670 Pionki, Poland; 2AdiFeed Sp. z o.o., Opaczewska, 02-201 Warszawa, Poland; 3Department of Food Chemistry and Biocatalysis, Wrocław University of Environmental and Life Sciences, C.K. Norwida 25, 50-375 Wrocław, Poland; 4Laboratory of Molecular Biology, Vet-Lab Brudzew, Turkowska 58c, 62-720 Brudzew, Poland; 5Laboratory of Industrial and Experimental Biology, Institute for Health and Economics, Carpathian State University in Krosno, Rynek 1, 38-400 Krosno, Poland

**Keywords:** *E. coli*, APEC, phytobiotics, antibiotic alternative, foodborne disease, antibiotic resistance, resistance genes, MIC, thymol, 1,8-cineole

## Abstract

*Escherichia coli* infections (including APEC) in broiler chickens are not only a health and economic problem of the flock, but also a significant health threat to poultry meat consumers. The prophylactic and therapeutic effects of the phytobiotic composition on *E. coli* in broiler chickens were previously described. However, most of the data were related to the reference strains (for both in vitro and in vivo models). Based on the previous studies in human and animals, *E. coli* strains seem to be multidrug resistance. This, in turn, makes it necessary to develop effective alternative methods of treating this type of infection already at the stage of poultry production. In the present study, the antibacterial activity against various strains of *E. coli* (including APEC) was assessed for two innovative phytobiotics mixtures: H1, containing thymol, menthol, linalool, *trans*-anethole, methyl salicylate, 1,8-cineol, and *p*-cymene; H2, in addition to compounds from H1, containing terpinen-4-ol and γ-terpinene. The unique mixtures of phytobiotics used in the experiment were effective against various strains of *E. coli*, also against APEC, isolated from broiler chickens from traditional industrial breeding, as well as against those showing colistin resistance. The minimum inhibitory concentration (MIC) values for these unique mixtures were: For H1 1:512 for APEC and non-APEC *E. coli* strains isolated from day old chicks (DOCs), 1:512 for non-APEC, and 1:1024 for non-APEC isolated from broilers sample. For mixture H2, MIC for APEC from both type of samples (DOCs and broilers) was 1:1024 and for non-APEC (DOCs and broilers) was 1:512. The results suggest that phytobiotic compositions used in this study can be successfully used as a natural alternative to antibiotics in the treatment of *E. coli* infections in broiler chickens. The promising results may be a crucial point for further analyses in broiler flocks exposed to *E. coli* infections and where it is necessary to reduce the level of antibiotics or completely eliminate them, thus reducing the risk of foodborne infections.

## 1. Introduction

In the case of poultry, the problem of *Escherichia coli* (*E. coli*) has a multifaceted dimension. In broiler chickens’ flocks, this bacterium can cause not only health, but also economic problems. At various stages of rearing, it may cause mortality, the need to slaughter birds showing symptoms of disease (selection), as well as a decline in weight gain, lower final weights, and problems with uniformity in terms of bird weight and size. This ultimately affects the quality of the raw material delivered to the slaughterhouse and to the next part of the supply chain.

*Escherichia coli* are mostly bacteria that live in the intestines of humans and animals without causing any harm to the host. Among the *E. coli*, those that can produce Shiga-toxin (STEC) and thus threaten human life have been identified. It should be noted that different strains of *E. coli* can contaminate water and different types of food [1].

Drug-resistant bacteria are also becoming a new danger for the consumer. Among the most important food-borne etiological factors related to poultry meat, the epidemiological reports mention: *Salmonella* spp., *Campylobacter* spp., *Verotoxic Escherichia coli* (VTEC), and *Shigatoxic E. coli* (STEC). Analyzing studies on poisoning in humans associated with *E. coli*, it can be noticed that the problem concerns both raw, frozen and heat-treated poultry products [2,3,4]. Moreover, poisoning occurs in highly developed and developing countries or in countries with a low development rate [5,6,7]. This, in turn, clearly indicates that the problem is global and requires wider analysis and implementation of solutions leading to a reduction in the number of infections at the level of broiler chicken flocks, as well as reducing the risk of contamination throughout the entire supply chain.

Recent studies have shown that many avian pathogenic *E. coli* (APEC) are now found in human isolates, which may increase the risk of a range of diseases and drug resistance. Moreover, it is worrying that the isolated strains show the ability not only to cause diseases of the gastrointestinal tract, but that APEC can also cause food-borne urinary tract infections (FUTIs) [8].

Poultry colibacillosis is a disease caused by pathogenic *Escherichia coli* strains, including Avian Pathogenic *Escherichia coli* (APEC). All species of birds of all ages are susceptible to infection, but the greatest losses are observed in chicks after hatching, broiler chickens. Colibacillosis in chicken broiler flocks can cause high mortality and negatively affect production parameters (high FCR), which ultimately causes huge economic losses at many stages of poultry production.

APEC strains, which include a large number of *E. coli* sequence types, are one of the subpathotypes of ExPEC (Extra Intestinal Pathogenic *E. coli*), which also include uropathogenic *E. coli* associated with sepsis and neonatal meningitis [9]. Many studies emphasize that APEC and human ExPEC strains have similar virulence-related genes, which may increase the risk of zoonotic *E.coli* infections in humans [10,11]. This causes an increasing demand for alternative antibacterial solutions that are highly effective, but also safe for consumers and allow for farming without antibiotics while maintaining at least the same good production parameters as in traditional farming with the use of antibiotics.

For years, special organizations have been analysing the phenomenon of resistance to *E. coli,* for both animals and humans, as well as for strains isolated from food. Previous studies have shown that resistance has been increasing in both clinical and food-borne *E. coli*, with the fastest an increase in resistance among animal isolates. [12]. Despite the increasing public health concerns over antibiotic use (both in human and veterinary medicine), the numbers related to their sale and distribution are still high, although during the last years (after the year 2016) they have decreased [13]. What is even more worrying is the fact that more and more strains show resistance to fluoroquinolones, cephalosporins, or the aforementioned polymyxins [14]. It is these groups of antibiotics that constitute the strongest line of defence against *E. coli* infections. What is also concerning is that many strains of *E. coli* isolated from humans and animals show Multidrug resistance (≥3 antimicrobial drug classes). The analysis presented by us are the first stage of work on a mixture of phytobiotics, which has the potential to be an alternative to classical antibiotic therapy in broiler flocks. In the next stage, it is planned to test different doses of the product in broiler flocks at the time of confirmation (necropsy + microbiological examination with antibiogram) of *E. coli* infection. This will provide a full answer to the question of whether the mixture developed and tested in the study is also effective in the in vivo environment and can replace popular antibiotics, contributing to the reduction of their use and increasing drug resistance.

Many scientific papers show that an alternative to antibiotics and antibiotic growth promoters could be plant-based or plant-derived products. The most promising results, against *E. coli*, have been obtained with the use of plant extracts and plant-derived compounds, such as polyphenols, alkaloids, or essential oils [15,16,17,18,19].

Finding alternatives to antibiotics is very important, not only because of the bans that have been applied, but also for antibiotic protection. As Gambi et al. demonstrate, eliminating the excessive use of antibiotics in broiler chickens rearing significantly reduces the presence of antibiotic resistance genes in bacteria isolated from chicken carcasses [20]. This contributes to a reduction of antimicrobial resistance (AMR), allowing antibiotics to be protected and used when necessary. Another example of antibiotic protection is combined therapies. These involve the use of, e.g., plant-derived compounds and antibiotics. Appropriate selection of active substances allows to achieve a synergistic effect, and thus greater effectiveness of treatment and the possibility of breaking antibiotic resistance [18,21,22,23]. 

Moreover, plant extracts, compounds, and essential oils can be combined with other similar fractions or with organic acids, fatty acids, probiotic bacteria, and metal ions, showing significantly better results in terms of either production parameters or antimicrobial properties [17,19,24,25,26,27,28]. Merging different compounds allows to reach the synergistic effect of the mixture and makes it much more difficult for bacteria to develop resistance.

The presented study is a continuation of previous analyses to evaluate the effectiveness of a mixture of phytobiotics against selected *Salmonella* strains isolated from field trials, as well as reference strains [29]. This study presents the effectiveness analysis against various strains of *E. coli* (including APEC) isolated from broiler flocks from industrial farms in Poland subject two mixtures of phytobiotics: (1) with exactly the same composition as the one presented in the previous study; (2) being a modification of the first in terms of the amount and type of ingredients, two more phytoncides were added to the mixture: terpinen-4-ol and γ-terpinene.

The aim of the present study was to determine the antibacterial activity, of phytobiotic mixtures containing thymol, menthol, linalool, *trans*-anethole, methyl salicylate, 1,8-cineole, *p*-cymene, terpinen-4-ol, and γ-terpinene, in vitro against different strains of *E. coli* (also APEC) isolated from different types of samples collected from commercial broiler farms.

It should be emphasized that the conducted analyses are important not only for veterinary medicine (due to the potential effective therapeutic effect of the preparation), but, indirectly, for food safety, due to the potential possibility of reducing the consumption of antibiotics and thus protecting their effects, as well as reducing the risk of residues or reducing the possibility of contaminating consumers by reducing the incidence of disease in the broiler flock from which the meat will go to food production.

## 2. Results

### 2.1. Anatomopathological Examination

The post-mortem lesions in broilers were acute, indicating generalized infection. In necropsy hepato- and splenomegaly, fibrinous peritonitis, pericarditis, and a thin layer of fibrous exudate located on liver and air sacs were present. Gross lesions in broilers were characteristic for generalized colibacillosis in 84% (50 out of 59), manifested by aerosacculitis (44%), fibrinous peritionitis (77%) and pericarditis (32%), hepatitis (54%), hepatomegaly (44%), splenomegaly (35%), and enteritis (26%), during the anatomopathological examination. In day old chicks, most frequent lesions were hyperemia of the yolks (79%) and navel area (69%), lung congestion (27%), and urate deposits in ureters (58%).

### 2.2. Identification of Isolates

All 92 isolates were correctly identified as *E. coli* using a commercial *E. coli* real-time PCR test (Genesig, PrimerDesign, Chandler’s Ford, Eastleigh, UK). Internal control was analyzed in the VIC-TAMRA channel, while the result for the specificity of *E. coli* identification was read in the FAM-TAMRA channel. The sample was considered positive if a characteristic amplification curve was visible in the FAM-TAMRA and VIC-TAMRA channels.

All 92 isolates were correctly identified as *E. coli* using APIE and VITEK2. One of strains isolated from 2-week-old broiler chickens was beta-hemolytic.

### 2.3. Somatic Antigen

Amongst *E. coli* isolates, 32 (29%) reacted positively with the sera used. The remaining 71% of the isolates did not show a positive reaction with the sera. The most frequently noticed serogroup was O1 (23%) and O2 (20%), followed by O78 (3%). Further, 54% of the tested strains were not typeable using O1, O2, or O78 sera.

### 2.4. Virulence Gene Detection

The PCR study showed that virulence genes were present as described in Table 1: *astA* (73 out of 92, 79.35%), *iss* (84 out of 92, 91.3%), *irp2* (66 out of 92, 71.74%), *papC* (29 out of 92, 31.52%), *cvi*/*cva* (45 out of 92, 48.91%), *iucD* (69 out of 92, 75%), *tsh* (34 out of 92, 36.96%), *vat* (39 out of 92, 42.39%), *iutA* (33 out of 92, 35.87%), and *ompT* (19 out of 92, 20.65%). *E. coli* harbored seven and more virulence gene was identified as APEC. According that definition, in our study, 39 of the 92 strains (42.39%) were defined as APEC. One *E. coli* APEC strain isolated from broiler harbored eight APEC-specific virulence genes (*astA*, *iss*, *irp2*, *papC*, *iucD*, *tsh*, *vat*, *iuaT*). The distribution of virulence genes in *E. coli* APEC and non-APEC is presented in Table 2, Figure 1 and Figure 2.

### 2.5. Prevalence of Multiple Drug Resistance

In our study, most of *E. coli* APEC and non-APEC demonstrated an MAR index lower than 0.3. One *E. coli* APEC isolate from day old chick showed an MAR index 0.32, three *E. coli* APEC isolates from broiler chickens showed an MAR index 0.32, and three isolates 0.36. Four *E. coli* non-APEC showed MAR index 0.32. MAR Index results are shown in Appendix A.

### 2.6. Antimicrobial Resistance Profile

All isolated *E. coli* were sensitive to colistin (COL) and none of the strains showed the presence of carbapenemase. One of APEC strain isolated from broilers was sensitive for all tested antimicrobial agents and harbored seven virulence genes (*astA*, *iss*, *irp2*, *cvi/cva*, *iucD*, *tsh*, *ompT*). In necropsy, hepato- and splenomegaly, fibrinous peritonitis, pericarditis, and a thin layer of fibrous exudate located on liver and air sacs were present. One *E. coli* APEC isolated from broiler was resistant only to cephapirin (I generation cephalosporin) and harboured seven virulence genes (*astA iss*, *irp2*, *cvi/cva*, *iucD*, *tsh*, *ompT*) or amoxicillin isolated from day old chicks (*astA iss*, *irp2*, *cvi/cva*, *iucD*, *vat*, *iutA*). Only one detected *E. coli* β (AMX-STR-FLR-LIN/SP-TR/SMX) was non-APEC and harbored four virulence genes (*astA*, *iss*, *iucD*, *tsh*). In necropsy, spleen and lung congestion, splenomegaly fibrinous peritonitis, pericarditis, and a thin layer of fibrous exudate located on liver and kidney swelling were present. *E. coli* APEC resistant for gentamycin isolated from DOCs AMX-GEN-STR-NOR-DOX-OXY-LIN/SP-TR/SMX and AMX-CPH-GEN-NOR-DOX-OXY-TR/SMX harbored seven virulence genes (*astA iss*, *irp2*, *cvi/cva*, *iucD*, *tsh*, *ompT*, and *astA iss*, *irp2*, *cvi/cva*, *iucD*, *tsh*, *iutA,* respectively). Results are shown in Appendix A.

### 2.7. Genotypic Resistance

The gene *blaCMY-2* encoding class C beta-lactamase same resistance to ceftiofur was not detected both in APEC and non-APEC strains. Thus, none of strains presents phenotypic resistance to III generation cephalosporins. In addition, none of the strains harbored genes *blaPSE-1* and *blaTEM* responsible for the ampicillin and penicillin resistance. The gene *blaSHV*, which confers resistance to cephalosporins and penicillins such as amoxicillin, was detected in all strains resistant to amoxicillin. The genes *aadA* (*ANT*(3″) and *strA/strB (APH(6)* both encoded by plasmids and associated with resistance to streptomycin were detected in 23 (25%) strains. The *tetA* and *tetB* genes, encoding tetacycline efflux pumps, conferring resistance to tetracycline, were detected in all strains resistant to doxycycline and oxytetracycline. Sulphonamide-resistant strains harbored at least one *sul* (1, 2, 3) and *adfR* gene, of which the s*ul1*, s*ul2*, and *sul3* were the most frequently noticed genes. d*frA1* encoding dihydrofolate reductase was present in four (4.35%) APEC strains, while *dfrA10* and *dfrA12* in two strains. The gene *floR*, encoding chloramphenicol exporter, was detected in all strains resistant to florfenicol. Genes *aphA1* and *aphA2* associated with resistance to neomycin (coding aminoglycoside 3′-phosphotransferase) were present only in seven (7.61%) *E. coli* non-APEC with phenotypic resistance.

The distribution of the antibiotic resistance, AMR, and virulence genes and the prevalence of the corresponding day-old chicks and broilers are shown in Appendix A.

### 2.8. Effectiveness of Phytoncides Composition

Both analysed compositions show very good antibacterial properties. The values of minimum inhibitory concentration (MIC) and minimum bactericidal concentration (MBC) were at the level 1:512–1:1024. For the composition H1, the effective dilution for most isolated strains was 1:512. Only for APEC isolated from broiler chickens was slightly better efficiency demonstrated; 1:1024. The second mixture (H2) showed very similar results with effectiveness at 1:512 for APEC strains and 1:1024 for non-APEC.

The MIC and MBC values against *Escherichia coli,* for both mixtures, were nearly identical. Unexpectedly, both compositions were more efficient to APEC strains than non-APEC. Analysed dilution of the mixtures are presented in Table 3.

## 3. Discussion

The results of the present study showed that both phytobiotic mixtures (H1 and H2) are effective against selected *E. coli* strains isolated from infected materials (day old chicks, older broiler etc.) in an in vitro model. To our knowledge, this is the first study where the bacteriocidial effects on *E. coli* (included APEC) of phytobiotic compositions containing H1-thymol, menthol, linalool, *trans*-anethole, methyl salicylate, 1,8-cineol, *p*-cymene, terpinen-4-ol, and γ-terpinene are presented. Moreover, this study is a continuation of previous observations, also in an in vitro model using *Salmonella* spp. strains from trials obtained on industrial farms. Current analyses together with previous results indicate that the mixtures of phytobiotics used in the study may be an alternative to classic antibiotics in the fight against bacterial infections of poultry responsible for foodborne disease in humans. In the case of effective action on *E. coli* strains (including APEC) isolated from material collected from industrial farms, this solution, after confirming its effectiveness in an in vitro model, can effectively improve the uniformity of the flock, reduce the occurrence of the mentioned infections, and indirectly improve the economy of poultry production at various stages and help reduce the growing drug resistance among bacteria that pose a threat to both human and animal health and life. What is observed on farms, but also reported in scientific publications, is the increasing drug resistance of *E. coli* strains including APEC [28,29,30]. Drugs routinely used to control this type of infection in poultry are either banned by recipients of livestock, or their administration requires a number of measures. In the light of such problems, the obtained results are extremely important and may be the starting point for the development of effective therapeutic solutions that will effectively treat these infections as well as limit the increasing drug resistance.

Among the flock where different *E. coli* strains were diagnosed and treated based on antimicrobial sensitivity test, during the pre- and post-mortem inspections made routinely in the slaughterhouse, most presented typical symptoms, such as airsacculitis, pericarditis, and perihepatitis (Figure 3). Moreover, numerous fibrous exudate deposits on other organs and body cavities were also observed. In addition, the uniformity of the flocks slaughtered shortly after diagnosis and application of classical antibiotic therapy were low (an exemplary diagram for such a herd is presented in Figure 4).

In the sensitivity test, 25 antibiotics were used, including: aminopenicillin (amoxicillin, amoxicillin and clavulanic acid), I generation cephalosporins, (cephalexin, cephapirin), III generation cephalosporins (ceftiofur), IV generation cephalosporins (cefquinome), penicillin cloxacillin, penicillin G, nafcillin), aminoglycoside (gentamicin, neomycin, streptomycin), polymyxins (colistin), fluorochinolones (enrofloxacin, norfloxacin), tetracyclines (doxycycline, oxytetracycline), macrolides (erythromycin, tylosin), florfenicol, lincosamides (lincomycin, lincomycin/spectinomycin), trimethoprim-sulfamethoxazole X, tiamulin, and tylvalosin.

As has been emphasized many times in previous studies [29,30,31], phytobiotics are an excellent alternative to classic antibiotic therapy in veterinary medicine, especially in food-providing animals. This is mainly due to the higher safety of this type of product (fewer side effects, lower toxicity, and, most importantly, no drug resistance and no withdrawal period). In this study several of *E. coli* presented high sensitivity against both phytobiotics mixtures. Moreover, APEC strains also showed sensitivity to the applied phytobiotic mixtures, which should be considered as a good prognostic sign due to the fact that APEC very often show high drug resistance.

Many scientists prove that compounds of natural origin, especially essential oils and compounds found in them, can be an effective alternative in the prevention and control of G-positive and G-negative bacteria. Unfortunately, the effective concentration is often characterized by a relatively high dose, thus impacting sensory qualities [32]. Therefore, it is becoming more common practice to combine different essential oils, or their components, to achieve a synergistic effect and improve their properties [33,34]. Combination with antibiotics also shows much better results with lower dosage [35].

The proposed composition shows very high activity against *Escherichia coli* strains. The high efficacy of the used formulation is the result of the composition containing active compounds showing at least an additive or even synergistic effect, rather than antagonism, as can occur in some cases [35,36,37]. The combinations studied in the literature consisted of one maximum of several active components, most often found in the same essential oil or belonging to the same group of chemical compounds. 

Methyl salicylate has a very high growth inhibitory activity against *E. coli*. Both the pure compound and the essential oil, in which it accounts for more than 90%, had very high growth inhibition zone [38]. The oil extracted from *Gaultheria procumbens L.*, which contains mainly methyl salicylate and limonene, showed similar properties. [39].

In our study, we observed that the combination (H2) containing additionally terpinen-4-ol and γ-terpinene shows slightly worse results against APEC from DOCs than the composition without them (H1). We did not observe the antagonistic effect between used compounds [36]. According to some literature reports, no synergistic effect was observed between carvacrol and thymol and their precursors p-cymene and γ-terpinene [40]. In another case, Soulaimani et al. also recorded no synergistic or additive effect against *E. coli* for essential oils rich in thymol, p-cymene, and γ-terpinene with an oil rich in 1,8-cyneol. In our studies, both compositions containing the aforementioned components showed strong antimicrobial activity, but no increase in activity against *E. coli* was observed for the H2 composition [34]. The slightly lower effectiveness of the H2 composition could be attributed to a reduction in the proportion of methyl salicylate and other components, for an increased amount of terpinen-4-ol and γ-terpinene. Salicylates show the ability to inhibit the growth of fimbriae, and thus the adhesion of *E. coli* to the epithelium and other surfaces. Moreover, the presence of salicylates increases the antimicrobial activity, especially that of aminoglycosides antibiotics. [41]. These reasons likely explain why the slightly better efficiency was characterized by the H1 mixture containing a higher amount of methyl salicylate.

## 4. Materials and Methods

### 4.1. Phytoncides Mixtures

Seven common phytoncides were selected for the tests: thymol, menthol, linalool, *trans*-anethole, methyl salicylate, 1,8-cineol, *p*-cymene, terpinen-4-ol, and γ-terpinene. All compounds were purchased from Sigma-Aldrich (St. Louis, MO, USA) complaint to FCC and FG standards. Purity and percentage composition, according to supplier specification, were minimum ≥95%.

All phytoncides were mixed in equivalent amounts, heated, and left overnight. The prepared mixture was then mixed with an emulsifier (Polysorbat 80, Sigma-Aldrich) for easier dissolution in aqueous solutions and culture media. Two mixtures were prepared for the further tests. The first one (H1), as well as in the previous test against *Salmonella* spp., contained thymol, menthol, linalool, *trans*-anethole, methyl salicylate, 1,8-cineol, and *p*-cymene [29]. The second mixture (H2), except compounds from the H1, also contained terpinen-4-ol and γ-terpinene.

### 4.2. Broiler Sampling

Samples were routinely collected from industrial broiler farms in Poland over 12 months (May 2021–May 2022). Samples were taken from day old chicks (birds were taken from the hatchery car cages, without any contact to the farm environment) and dead birds from flocks that were suspected of having colibacillosis (clinical symptoms and routine autopsy on the farm made by the local veterinary company). Birds were taken for testing in a situation of increasing/high mortality, an increased number of selection birds (small, lame birds, etc.), as well as a decrease in weight and feed/water intake. Freshly dead birds and selected birds showing signs of disease (e.g., asphyxiation, lower weight, mobility problems, lameness, ruffled feathers etc.) were selected for the post-mortem examination. Birds were delivered to the laboratory for microbiological examination at refrigeration temperature. For best results, dead animals were necropsied during 2 h after death. Birds that have been dead for more than a few hours are not recommended for diagnostic specimens since the natural autolytic process will create post-mortem lesions that may be mistake with true pathological lesions. The standard autopsy procedure includes: external and internal examination with necropsy description. External examination includes inspection of feathers (loss, ruffling) and/or nasal or ocular discharge. After cutting and removal of the skin over the abdomen, a breast muscle for decreased muscle mass was examined. Next, internal organs and chest cavity were exposed. The liver and spleen were examined for changes in size or discoloration, white or yellow spots, abscesses, and fibrinous exudate. The air sacs were examined for increased thickness and increased cloudiness. Next, proventriculus, gizzard, small intestines, large intestine, and ceca were checked. Subsequently, lungs, outer surface of the heart, muscles and valves were examined. Finally, hock and stifle joints and tendon sheath were examined. Organ sampling was performed during necropsy according to an internal laboratory protocol.

The study was performed on 92 different *E. coli* isolated strains (33-day-old chicks, 59 broiler chickens).

### 4.3. Escherichia coli Isolation and Identification

Samples (liver, spleen, lungs, air sac, joints, and spinal cord) were taken from DOCs and broilers from different flocks using a sterile inoculation loop (organs) or swabs and directly inoculated on Columbia Agar with 5% defibrinated sheep blood and MacConkey agar (both from Graso, Starogard Gdański, Poland). Plates were then incubated at 37 °C for 24 h under aerobic conditions.

*E. coli* isolates were identified based on colony morphology and lactose fermentation. Isolates initially identified as *E. coli* were confirmed using a commercial real-time PCR Genesig (PrimerDesign, Chandler’s Ford, Eastleigh, UK) and Applied 7500FAST (ThermoFisher, Waltham, MA, USA). The *Escherichia coli* ATCC 25922 and *Enterococcus faecalis* ATCC 29212 reference strains were used as positive and negative controls. Nuclease-free water was used as a second negative control.

Identification of *E. coli* was additionally confirmed by biochemical identification using two commercially available tests: API 20E (BioMérieux, Craponne, France) and a VITEK2 COMPACT with VITEK^®^2 GN cards (BioMérieux, Craponne, France). Reference strain *E. coli* ATCC 25922 served as a quality control. Both tests were used according to the manufacturer’s instructions.

### 4.4. Somatic Antigen Identification

*E. coli* isolates were next seeded onto Nutrient Agar (OXOID, Hampshire, UK) for serotyping.

All *E. coli* strains were tested by slide agglutination against antigens: O1, O2, O78 (Sifin, Berlin, Germany), which are the most common in poultry.

### 4.5. APEC Genes Detection

DNA for PCR was isolated from *E. coli* cultures using a fully automated nucleic acid extraction system (AutoPure96, Wuxi, China). All 92 isolates were analyzed for the presence of eight virulence genes (*vat*, *tsh*, *iucD*, *cvi*/*cva, papC, irp2, iss*, and *astA)* and by an end-point PCR Kylt^®^ APEC kit (AniCon Labor GmbH, Höltinghausen, Germany) according to manufacturer instructions. The PCR product bands were visualized under ultraviolet light using automated UV gel documentation system (UVP Solo, Analytic Jena, Jena, Germany) after electrophoresis on a 2% agarose gel with ethidium bromide. The PCR reaction products were interpreted by comparing them to size standards (2+ Marker, A&A Biotechnology, Gdańsk, Poland) and positive controls, according to the Kylt^®^ APEC manual of instructions as described in Table 4. Additionally, real-time EXOone APEC kit (EXOPOL, Zaragoza, Spain) was used to detect two virulence genes (*iutA*, *ompT*) according to manufacturer instructions. Real-time PCR was performed using Applied FAST 7500 (ThermoFisher, Waltham, MA, USA). *E. coli* harbored seven and more virulence gene was identified as APEC.

### 4.6. Antimicrobial Susceptibility of Escherichia coli Isolates

Each *E. coli* strain was first subcultured as described previously. Antimicrobial susceptibility was evaluated by determining the MIC values using a 96-well MICRONAUT Special Plates with antimicrobials: aminopenicillin (amoxicillin, amoxicillin and clavulanic acid), I generation cephalosporins, (cephalexin, cephapirin), III generation cephalosporins (ceftiofur), V generation cephalosporins (cefquinome), penicillin cloxacillin, penicillin G, nafcillin), aminoglycoside (gentamicin, neomycin, streptomycin), polymyxins (colistin), fluorochinolones (enrofloxacin, norfloxacin), tetracyclines (doxycycline, oxytetracycline), macrolides (erythromycin, tylosin), florfenicol, lincosamides (lincomycin, lincomycin/spectinomycin), trimethoprim-sulfamethoxazole, tiamulin, tylvalosin (MERLIN Diagnostika GmbH, Bremen, Germany). Additionally, all strains were screened for the presence of carbapenemase using RAPIDEC^®^ CARBA NP (BioMérieux, Craponne, France). Reference strain *E. coli* ATCC 25922, *S. aureus* ATCC 25923, and *P. aeruginosa* ATCC 27853 were used as negative controls. The MICs were interpreted in accordance with Clinical and Laboratory Standards Institute (CLSI) and FDA breakpoints (CLSI M100-ED28, 2018). For calculation of multiple antibiotics resistance, we used the formula of Akinola et al. [42].
$$\mathrm{MAR}=\frac{\mathrm{Number}\;\mathrm{of}\;\mathrm{resistance}\;\mathrm{to}\;\mathrm{antibiotics}}{\mathrm{Total}\;\mathrm{number}\;\mathrm{of}\;\mathrm{antibiotics}\;\mathrm{tested}}$$

### 4.7. Detection of Antibiotic Resistance Genes

Bacterial DNA isolation was performed as described above from overnight bacterial culture on nutrient agar at 37 °C. Hence, 19 resistance genes (*aadA*, *strA/strB*, *aphA1*, *aphA2*, *aadB*, *tetA*, *tetB*, *sul1*, *sul2*, *sul3*, *dfrA1*, *dfrA10*, *dfrA12*, *floR*, *blaTE*M, *blaSHV*, *blaCMY*-2, *blaPSE-1,* and *blaCTX-M*) were studied by end-point PCR, using multiplex PCR or a single PCR reaction. The primer sequences, PCR product sizes, and references are shown in Appendix A (based on [43,44]).

### 4.8. Phytoncides Mixture Test by Broth Microdilution Method

The H1 and H2 mixture were evaluated in this study. The antimicrobial activity of the phytoncides mixture was tested using the broth microdilution method described in ISO 20776-1:2006. In sterile vials, two-fold serial dilutions of the phytoncides mixture were prepared in Mueller Hinton II Broth (M-H Broth) with a final volume of 2 mL per vial. The inoculum was prepared in sterile 0.9% NaCl solution, and it was derived from the overnight culture of each bacteria isolate on sheep blood agar, adjusting the turbidity to 0.5 McFarland standard. Subsequently, the suspensions were diluted a hundredfold in M-H Broth by transferring 110 µL of the suspension into 11 mL M-H Broth to obtain 106 CFU/mL of inoculum. Next, 1 mL of bacterial inoculum was transferred into vials containing 1 mL of diluted mixture, resulting in the following test dilutions, per row: 4, 8, 16, 32, 64, 128, 256, 512, 1024, 2048, 4096, and 8192. Vials containing 1 mL of M-H broth only, without product, and 1 mL of inoculum were used as positive growth controls. Wells containing 1 mL of diluted product (a two-fold dilution series) and 1 mL of M-H broth without any of the bacterial isolates were used as negative controls. Finally, vials were incubated at 35 ± 1 °C for 21 ± 3 h. After incubation, the lowest concentration (the highest dilution) of the product that completely inhibits visible growth was recorded. The minimum inhibitory concentration (MIC). To check for purity, after inoculation of the vials, bacterial suspensions made in saline were streaked onto Columbia Agar with 5% Sheep Blood agar. Following overnight incubation at 37 °C, cultures were checked for morphologically characteristic colonies.

## 5. Conclusions

The results presented in this study show the antibacterial activity of two mixtures of phytobiotics against *E. coli* strains (included APEC) isolated from material collected from day old chicks and older broiler from commercial poultry farms. The study is a continuation of the observations carried out on various strains of *Salmonella* spp. [29] with the difference that, in the current study, a second, compositionally modified mixture of phytobiotics was used, which also showed high effectiveness against *E. coli* including APEC. The obtained results complement previous analyses and provide new, promising data on the possibility of reducing and controlling bacteria in broiler flocks responsible for foodborne disease in humans. Previous studies conducted on an in vivo model on broiler chickens have shown that phytobiotic mixtures can be effective alternatives to antibiotic growth promoters, improving selected production parameters, but also affecting myokines and interleukins related to immunity and muscle growth [45]. The two mixtures tested in the presented study, after confirming their effectiveness on an in vitro model, could be a perfect complement to a comprehensive antibiotic reduction program in poultry production with the use of various alternatives to classic antibiotics, including other phytobiotic mixtures.

However, it should be noticed that many times the results obtained in the in vitro environment may be different from what is observed in real conditions on factory farms. For this reason, the study should be considered a preliminary test that will require verification in vivo.

## Figures and Tables

**Figure 1 antibiotics-11-01818-f001:**
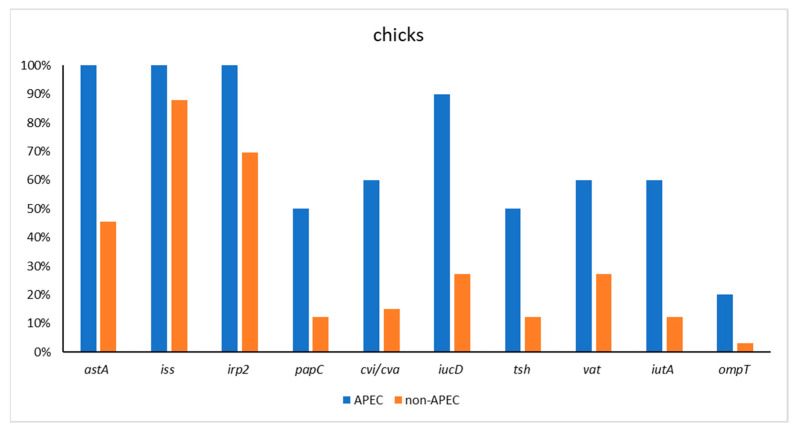
Virulence gene distribution *E. coli* APEC and non-APEC strains in day old chicks.

**Figure 2 antibiotics-11-01818-f002:**
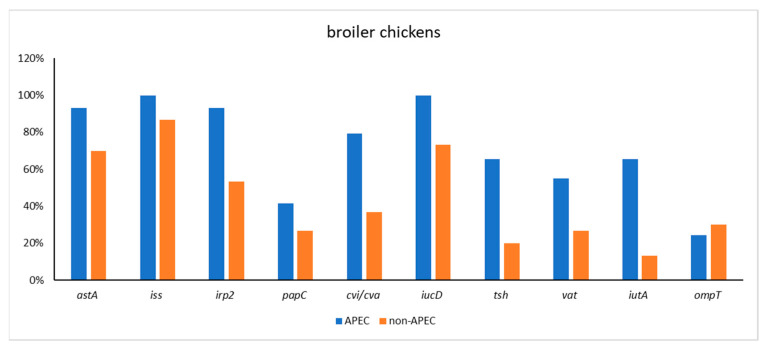
Virulence gene distribution *E. coli* APEC and non-APEC strains in broiler chickens.

**Figure 3 antibiotics-11-01818-f003:**
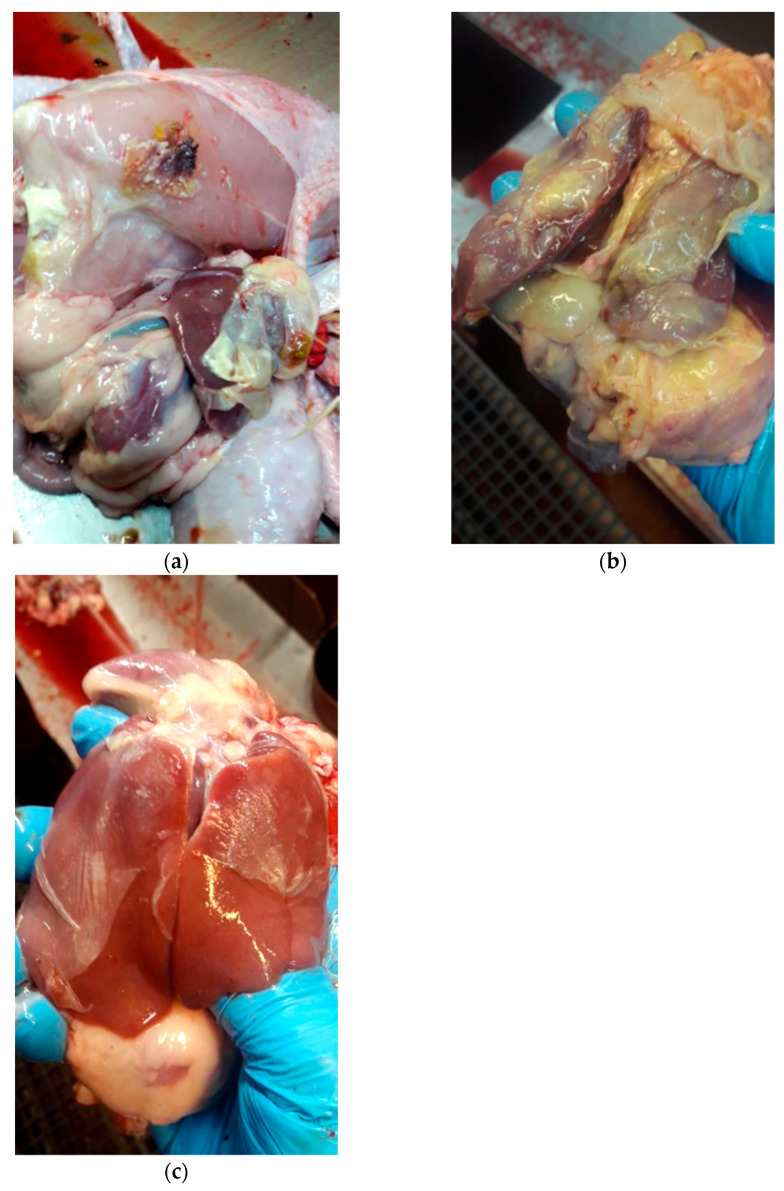
Postmortem examination of the flocks where *E. coli* was previously diagnosed (including APEC). Changes were present on both; inside the carcass and organs; fibrinous peritonitis, pericarditis and thin layer of fibrous exudate located on liver (**a**–**c**). In addition to the changes described above, there was a visible flock uniformity in terms of weight and size of birds delivered to the slaughterhouse.

**Figure 4 antibiotics-11-01818-f004:**
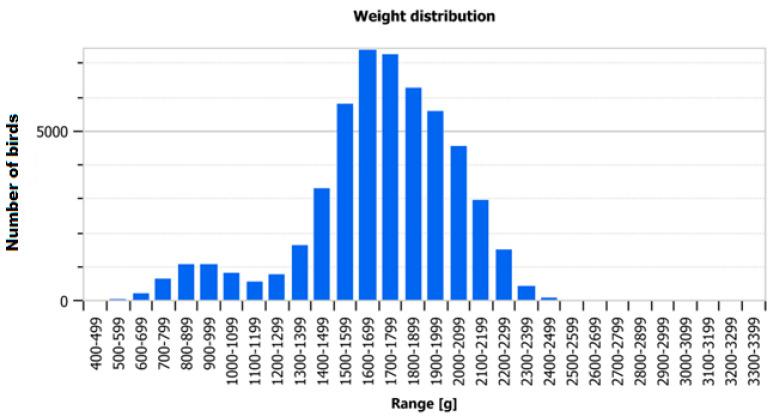
Several flocks where *E. coli* infection was diagnosed presented a large variation in carcass size and weight.

**Table 1 antibiotics-11-01818-t001:** Distribution of virulence genes in APEC and non-APEC strains.

*E. coli* Strain		
Genes	Negative *n* (%)	Positive *n* (%)
*astA*	19 (20.65)	73 (79.35)
*iss*	8 (8.69)	84 (91.31)
*irp2*	26 (28.26)	66 (71.74)
*papC*	63 (68.48)	29 (31.52)
*cvi*/*cva*	47 (58.09)	45 (48.91)
*iucD*	23 (25.00)	69 (75.00)
*tsh*	58 (63.04)	34 (36.96)
*vat*	53 (57.61)	39 (42.39)
*iutA*	59 (64.13)	33 (35.87)
*ompT*	73 (73.35)	19 (20.65)

**Table 2 antibiotics-11-01818-t002:** Percentage of virulence genes in tested *E. coli* APEC and non-APEC strains.

Type of Poultry	*astA* *n* (%)	*Iss* *n* (%)	*irp2* *n* (%)	*papC* *n* (%)	*cvi*/*cva* *n* (%)	*iucD* *n* (%)	*tsh* *n* (%)	*Vat* *n* (%)	*iutA* *n* (%)	*ompT* *n* (%)
DOCs APEC (*n* = 10)	10 (100)	10 (100)	10 (100)	5 (50)	6 (60)	9 (90)	5 (50)	6 (60)	6 (60)	2 (20)
DOCs non-APEC (*n* = 23)	15 (45.45)	29 (87.88)	23 (69.7)	4 (12.12)	5 (15.15)	9 (27.27)	4 (12.12)	9 (27.27)	4 (12.12)	1 (3.03)
broiler chicken APEC (*n* = 29)	27 (93.10)	29 (100)	27 (93.10)	12 (41.38)	23 (79.31)	29 (100)	19 (65.52)	16 (55.17)	19 (65.52)	7 (24.14)
broiler chicken non-APEC (*n* = 30)	21 (70)	26 (86.67)	16 (53.33)	8 (26.67)	11 (36.67)	22 (73.33)	6 (20)	8 (26.67)	4 (13.33)	9 (30)

**Table 3 antibiotics-11-01818-t003:** MIC of analyzed mixture.

***Escherichia coli* Strains**	**Dilution H1**											
	**1:2**	**1:4**	**1:16**	**1:32**	**1:64**	**1:118**	**1:256**	**1:512**	**1:1024**	**1:2048**	**1:4096**	**1:8192**
DOCs APEC	−	−	−	−	−	−	−	+	+	+	+	+
DOCs non-APEC	−	−	−	−	−	−	−	+	+	+	+	+
broiler chicken APEC	−	−	−	−	−	−	−	−	+	+	+	+
broiler chicken non-APEC	−	−	−	−	−	−	−	+	+	+	+	+
	**Dilution H2**											
	**1:2**	**1:4**	**1:16**	**1:32**	**1:64**	**1:118**	**1:256**	**1:512**	**1:1024**	**1:2048**	**1:4096**	**1:8192**
DOCs APEC	−	−	−	−	−	−	−	−	+	+	+	+
DOCs non-APEC	−	−	−	−	−	−	−	+	+	+	+	+
broiler chicken APEC	−	−	−	−	−	−	−	−	+	+	+	+
broiler chicken non-APEC	−	−	−	−	−	−	−	+	+	+	+	+

**Table 4 antibiotics-11-01818-t004:** The expected product sizes of the toxin genes of APEC.

No. of Band in Positive Control	Toxin Gene	Expected PCR Product Size
1	*vat*	978 bp
2	*tsh*	824 bp
3	*iucD*	693 bp
4	*cvi/cva*	598 bp
5	*papC*	501bp
6	*irp2*	413 bp
7	*iss*	309 bp
8	*astA*	111 bp

## Data Availability

Not applicable.

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
