# Peer review of "In Vitro Assessment of Antimicrobial Activity of Phytobiotics Composition towards of Avian Pathogenic Escherichia coli (APEC) and Other E. coli Strains Isolated from Broiler Chickens"

_antibiotics, 2022, doi:10.3390/antibiotics11121818_

Round 1

Reviewer 1 Report

Comments to authors:

L19: chicken broiler -> broiler chickens. Please also correct throughout the manuscript! There in inconsistency in using the term, the correct term is broiler chickens!

L36: what do you mean chicks and broiler?

Introduction

I am not certain if it is necessary to make subsections for the introduction, but I suggest to be drastically shortened the introduction. In the present form, it is too excessive and to much literature review, which is not effective to briefly explain your hypothetical development to the readers.

L169-170: it would be more relevant to put this reference https://doi.org/10.5713/ab.20.0668 and maybe other ref when mentioning the effect of essential oils on production parameters and E coli in broiler chickens.

Results

L186: please briefly explain the survey procedure before directly point to post mortem lesions etc, readers need to know a brief background in doing this.

Fig 1 and Fig 2: what do you mean to differentiate between chicks and broiler chickens?

Fig 4: pieces -> frequency/ number of birds?

L333: Thus impacting sensory qualities

L369: per-centage -> percentage

L384: chics -> DOC?

L387: flocks -> organs?

How you determine the dose of the mixture?

Author Response

Dear Reviewer please find our reply in the attached file.

Reviewer 2 Report

The authors conducted a study to evaluate the antibacterial activity of some phytochemicals against the avian pathogenic E.coli isolated from poultry farms. The topic is good and of interest due to the economic significance of the E.coli strains in poultry. Though the research design is executed correctly and results are clearly presented, the introduction section needs attention. The introduction is not only too large but lacks focus on the topic. The authors need to revise the introduction with adequate references that presents the background of this study.

Specific comments

1. If possible provide a simple title and include all details in the abstract

2. Page 11, broiler sampling: provide correct year or modify that statement

Author Response

(The authors gave the same response as above.)

Round 2

Reviewer 2 Report

The revised version is better than the last draft. Introduction section seems to be good.